# Immunotherapeutic Potential of Mutated NPM1 for the Treatment of Acute Myeloid Leukemia

**DOI:** 10.3390/cancers16203443

**Published:** 2024-10-10

**Authors:** Jochen Greiner, Eithar Mohamed, Daniel M. Fletcher, Patrick J. Schuler, Hubert Schrezenmeier, Marlies Götz, Barbara-ann Guinn

**Affiliations:** 1Department of Internal Medicine III, University Hospital Ulm, 89081 Ulm, Germany; marlies.goetz@alumni.uni-ulm.de; 2Department of Internal Medicine, Diakonie Hospital Stuttgart, 70176 Stuttgart, Germany; 3Centre for Biomedicine, Hull York Medical School, University of Hull, Hull HU6 7RX, UK; e.a.mohamed-2019@hull.ac.uk (E.M.); d.m.fletcher-2018@hull.ac.uk (D.M.F.); 4Department of Otorhinolaryngology, University Hospital Ulm, 89075 Ulm, Germany; patrick.schuler@uniklinik-ulm.de; 5Department of Oto-Rhino-Laryngology, University Hospital Heidelberg, 69120 Heidelberg, Germany; 6Institute of Transfusion Medicine, University of Ulm, 89073 Ulm, Germany; h.schrezenmeier@blutspende.de; 7Institute of Clinical Transfusion Medicine and Immunogenetics Ulm, 89081 Ulm, Germany

**Keywords:** acute myeloid leukemia, NPM1 mutation, immunogenic subtype, programmed death ligand 1, immunotherapy

## Abstract

**Simple Summary:**

One third of all patients with acute myeloid leukaemia (AML) carry a nucleophosmin mutation (NPM^mut^). This is associated with improved survival and which may be due to the immune response these cells can generate. However the reason is not fully understood. In this review we discuss the ways that patients with an NPM1^mut^ are treated clinically and how preclinical studies are informing best practice for patients with AML NPM^mut^.

**Abstract:**

Acute myeloid leukemia (AML) is a malignant disease of the blood and bone marrow that is characterized by uncontrolled clonal proliferation of abnormal myeloid progenitor cells. Nucleophosmin 1 (NPM1) gene mutations are the most common genetic abnormality in AML, detectable in blast cells from about one-third of adults with AML. AML NPM1^mut^ is recognized as a separate entity in the World Health Organization classification of AML. Clinical and survival data suggest that patients with this form of AML often have a more favorable prognosis, which may be due to the immunogenicity created by the mutations in the NPM1 protein. Consequently, AML with NPM1^mut^ can be considered an immunogenic subtype of AML. However, the underlying mechanisms of this immunogenicity and associated favorable survival outcomes need to be further investigated. Immune checkpoint molecules, such as the programmed cell death-1 (PD-1) protein and its ligand, PD-L1, play important roles in leukemogenesis through their maintenance of an immunosuppressive tumor microenvironment. Preclinical trials have shown that the use of PD-1/PD-L1 checkpoint inhibitors in solid tumors and lymphoma work best in novel therapy combinations. Patients with AML NPM1^mut^ may be better suited to immunogenic strategies that are based on the inhibition of the PD-1 immune checkpoint pathway than patients without this mutation, suggesting the genetic landscape of patients may also inform best practice for the use of PD-1 inhibitors.

## 1. Introduction

Nucleophosmin 1 (NPM1.1 or NPM1) is a mainly nucleolar protein that shuttles between nucleoli, nucleoplasm and cytoplasm chaperoning nucleic acids and proteins. It is expressed abundantly in all tissues [1] and plays a crucial role in cell cycle control, centrosome duplication, ribosome maturation and export, and cellular responses to a variety of stress signals. NPM1 is altered by overexpression, chromosomal translocations and mutations in solid and hematological cancers (reviewed in [2]). In hematological malignancies, the NPM1 gene is frequently combined with other genes to generate fusion proteins that retain the oligomerization domain in the N-terminus of NPM1, including NPM1-RARα in a subset of acute promyelocytic leukemia patients [3] and the rare NPM1-MLF1 [4] translocation, which is associated with the progression of myelodysplastic syndrome (MDS) to acute myeloid leukemia (AML).

The majority of point mutations in NPM occur in Exon 12 and are named alphabetically. Mutation A, a duplication of TCTG, is the most common [5], followed by Mutations B and D, and each of these point mutations, although different, almost always leads to the same outcome, a frameshift that creates a C-terminus sequence with a disrupted nucleolar localization signal and a third nuclear export sequence. As a consequence, NPM1 no longer predominantly resides in the nucleolus but relocates to the cytoplasm [5,6,7,8], where it is referred to as NPMc+. Accumulation of NPMc+ in the cytoplasm can be detected by immunocytochemistry [9] and qPCR, which allows the monitoring of molecular minimal residual disease (MRD) in patients [10]. NPM1c+ interacts with a number of other proteins (Figure 1), including itself, mostly using its N-terminal domain (reviewed in [11]).

Alterations in NPM1 are considered to be a gatekeeper mutation, one of the first hits in the process of leukemogenesis (reviewed in [12]). AML patients with a nucleophosmin-1 mutation (NPM1^mut^) are recognized as a separate entity by the World Health Organization (WHO) [12,13], since NPM1^mut^ is a founder genetic event that is rare in MDS [14] with distinct genetic, pathological, immunophenotypic, and clinical characteristics (Table 1). In this review, we will discuss AML NPM1^mut^ in the context of immunotherapeutic approaches that have been developed to treat it.

## 2. AML NPM1^mut^ and Prognosis

Survival rates for patients with AML are unfortunately still low. The 7 + 3 regimen achieves an estimated 5-year survival rate of 30% to 35% in younger adult patients (age < 60) and 10–15% in older patients (age > 60) [17]. Therefore, exploring new therapeutic options, especially for older patients, is crucial to improve survival rates and address risk factors. NPM1^mut^ are found more commonly in older adult AML patients (over 35 years of age) (Table 1) and correlate with high white cell counts and normal karyotypes [17]. They are present in the M1–M6 subtypes (French American British (FAB) classification of AML), but absent in AML M0 and patients with t(8;21), inv(16), t(15;17), or CCAAT/enhancer-binding protein-α (CEBPA) mutations [15]. Homeobox (HOX) genes (A4, A5, A6, A7, A9, A10, B2, B3, B5, B6) as well as the HOX-related genes, PBX3 and MEIS1, are upregulated in AML NPM1^mut^ samples, while CD34 mRNA levels are low/absent in AML patients with NPM1^mut^.

There are distinct patterns of co-mutations in NPM1^mut^ patients, and there are gene–gene interactions that are particularly pronounced, causing patterns of co-incidental mutationally defined groups with a favorable, or, conversely, adverse prognosis [18]. The concurrence of NPM1^mut^ is twice as frequent in patients with FLT3-ITD than those with NPM1^WT^. This may be explained by both being caused by replication slippage errors [19], with the belief that NPM1^mut^ precedes FLT3-ITD [5,20]. FLT3-ITD is associated with poor survival in younger (<60 years) but not in older (60–74 years) AML patients, while NPM1^mut^ is associated with good survival in older but not younger patients, when considering age as well as sex [21]. The European LeukemiaNet (ELN) 2022 results showed that younger patients with NPM1^mut^, who were FLT3-ITD negative and received high-dose chemotherapy, had a favorable risk classification as an initial risk assignment.

Ostronoff et al. [22] used a different age range and associated this genotype with a comparatively favorable prognostic factor for younger AML patients (55–65 years) but not for older patients (>65 years). Patients with NPM1mut but without a FLT3-ITD translocation often have a favorable prognosis [15], including a better overall survival (OS) rate [23].

Angenendt and colleagues [24] conducted a study on a group of over 2400 AML patients with NPM1^mut^/FLT3-ITD^neg/low^, who had varying karyotypes and chromosomal abnormalities. The results showed that in patients with NPM1^mut^/FLT3-ITD^neg/low^ AML, unfavorable cytogenetics were linked to a lower 5-year OS rate (52.4%, 44.8%, and 19.5% for normal, aberrant intermediate, and adverse karyotypes, respectively). This highlights the significant impact of karyotype abnormalities on the survival outcome of NPM1^mut^/FLT3-ITD^neg/low^ AML patients. In patients who have adverse-risk cytogenetics, those with NPM1^mut^ have a similarly poor prognosis as those with NPM1^WT^ and should receive treatment accordingly.

Over a long-term follow-up period of 60 months, survival, outcome, and the likelihood of relapse were similar in AML patients with and without NPM1^mut^, and this is because NPM1^mut^ are significantly associated with normal karyotypes and FLT3-ITD mutations (Table 2). In the group with an intermediate cytogenetic risk without FLT3-ITD but with NPM1^mut^, patients have a significantly better OS and EFS than patients without NPM1 mutations (*p* = 0.05) [15].

## 3. AML NPM1^mut^ Treatment Strategies

According to the ELN guidelines [12], new combination trials in patients eligible for intensive chemotherapy typically involve the incorporation of a new target or agent in combination with standard “7 + 3” chemotherapy. Examples of new targets or agents include ongoing clinical trials involving the FLT3 inhibitor gilteritinib versus midostaurin or the spleen tyrosine kinase (SYK) inhibitor for AML NPM1^mut^. In a randomized study, treatment with gemtuzumab–ozogamicin (GO) led to a reduction in the relapse probability and greater NPM1^mut^ molecular clearance but with no difference in EFS [35,36].

The NPM1^mut^ protein is a desirable target structure for customized immunotherapy approaches, especially for patients with chronic MRD or AML NPM1^mut^ patients receiving maintenance treatment. In an AML NPM1^mut^ patient in molecular relapse, pre-emptive DLI resulted in poly-specific cytotoxic T-lymphocyte (CTL) responses against multiple leukemia-associated antigens (LAAs), including NPM1 #3 [37]. Jäger et al. [38] examined 64 patients with AML NPM1^mut^ and showed that MRD-negative patients in complete remission (CR) achieve better 2-year progression-free survival (PFS) and 2-year OS compared with patients with MRD-positive patients in CR. They showed that alloHSCT in AML NPM1^mut^ patients depends on the disease burden, relapse type, and response to CT. AlloHSCT transplantation does not seem to benefit NPM1^mut^/FLT3-ITD-negative patients when used as a first-line treatment; nevertheless, further clinical trials are being conducted, and MRD needs to be considered [38].

AML NPM^mut^ cells are responsive to different cytotoxic agents such as demethylating agents and those that cause BCL-2 inhibition. DiNardo et al. [39] showed that the BCL2 inhibitor, venetoclax, achieved high response rates and durable remissions in older AML patients with NPM1 or IDH2 mutations. Subsequently, Jäger et al. [38] showed, albeit in a small number of patients, that hypomethylating agents combined with venetoclax may be an effective alternative treatment for AML NPM1^mut^, especially where an isocitrate dehydrogenase (NADP^+^ 2) IDH2 mutation is present. Effective relapse treatments were offered, with a particular emphasis on the incidence of mild or moderate graft versus host disease (GvHD), which is most likely caused by an immunological graft versus leukemia (GvL) effect.

Forghieri et al. [40] recently wrote about a promising therapeutic approach involving the use of neoantigen-specific T cells, i.e., genetically engineered T cells, such as CAR-T or TCR-transduced T cells directed against NPM1^mut^ peptides presented on HLA. In patients with full-blown leukemia, adoptive or vaccine-based immunotherapies may not be very effective, but these strategies, possibly in combination with immune checkpoint inhibitors (ICIs), may be able to maintain remission or pre-emptively eradicate persistent measurable residual disease (discussed further in Section 5.3). This also applies to patients who are not eligible for alloHSCT.

In summary, the main treatment for AML NPM1^mut^ in younger adults involves standard induction chemotherapy (7+3) with or without FLT3 inhibitors and consolidation rounds using high/medium doses of cytarabine. Allogeneic stem cell transplants may be used in first CR, based on FLT3 gene status and measurable residual disease levels. Recently, several new drugs have been approved for AML NPM1^mut^, including FLT3 and BCL-2 inhibitors. Multiple studies back the use of GO in the initial treatment of AML NPM1^mut^ [12,41]. Additionally, menin inhibitors constitute a promising new therapy for AML NPM1^mut^ [42].

## 4. Immunogenic Mutation-Related Targets

Liso et al. [43] demonstrated that mutant NPM1 peptides can be presented in the context of HLA, and showed that AML NPM1^mut^ cells could stimulate anti-leukemic T cell responses. Different CD4^+^ and CD8^+^ T cell responses to NPM1^mut^ epitopes have been identified [44], including two HLA-A2-restricted peptides, #1 and #3, which induced distinct T cell responses (33% and 44%, respectively). Compared with healthy individuals, AML NPM1^mut^ patients exhibited significantly higher frequencies of CTL responses against Peptide #3 (*p* = 0.046). In a survival analysis of 25 patients with AML NPM1^mut^, Greiner et al. [45] observed that those exhibiting specific CD8^+^ cytotoxic T cell immune responses against one or two immunogenic peptides had a better OS compared with those without such immune responses (*p* = 0.004). Schneider et al. [46] isolated a leukemic stem cells (LSC)-enriched population from both NPM1^mut^ and WT patients, and noted that the frequencies of CD34^+^CD38^−^ cells in AML NPM1^mut^ samples were significantly lower. They found a number of differentially expressed genes of immunological relevance in these populations, including immunoglobulin superfamily member 10 (*p* = 0.00034), CD96 (*p* = 0.00052) and IL-12 receptor beta 1 (IL12RB1, *p* = 0.000834). These markers are involved in immune mechanisms and antigen presentation, highlighting the potential of immune-based therapies that target this subtype of AML patients to be effective.

Kuželová et al. [47] found that in a cohort of 63 AML patients with NPM1^mut^, compared with 94 patients with NPM1^WT^, there was a significant decrease in HLA-B*07, B*18, and B*40 expression. There was also a notable OS advantage for AML NPM1^mut^ patients who expressed one of these depleted haplotypes (*p* = 0.02), suggesting that HLA-B*07, B*18 and B*40 are more effective at generating an anti-leukemia immune response.

Van der Lee et al. [48] identified multiple NPM1^mut^-derived peptides and demonstrated specific recognition and lysis of AML NPM1^mut^ cells after retroviral transfer to CD8+ and CD4+ T cells. The discovery of common peptide targets in AML NPM1^mut^ patients’ mutations provides an important immunotherapeutic treatment avenue for this subgroup of patients [49].

Among the most prevalent LAAs in AML are Wilms’ tumor antigen 1 (WT-1), Proteinase 3, and receptor for hyaluronan-mediated motility (RHAMM; reviewed in [50]). Cancer-testis antigens, such as helicase antigen (HAGE) and Per ARNT Sim domain containing 1 (PASD1), are expressed less frequently (23% and 35%, respectively) but have disease-specific expression [51,52], with little to no expression in healthy tissues, except in immunologically protected sites. All have been shown to generate antigen-specific CD8^+^ T cell responses from AML patients [53], and peptide vaccination studies investigating these antigens [54,55,56,57,58] revealed immunological and clinically relevant responses. Greiner et al. [59] examined anti-LAA responses using T cells from NPM1^mut^ and NPM1^WT^ AML patients. The authors showed that the anti-leukemia response was effective against NPM1^mut^ cells when the immunogenic epitope was derived from the mutated region of NPM1 compared with other epitopes/LAA, and these effects were enhanced through the addition of anti-programmed cell death protein 1 (PD-1).

## 5. Monoclonal Antibody Therapies

The standard treatment for AML has remained largely unchanged during the past few decades, which is the standard “7 + 3” regimen and includes a seven-day infusion of cytarabine and a three-day treatment with an anthracycline. It is commonly used to achieve CR and is particularly successful in patients aged 65 years and under. In recent years, significant advances in our understanding of the pathogenesis of AML, the development of diagnostic assays, and the approval of new treatments has led to updates in the standard care for AML patients.

### 5.1. αCD33

CD33 is primarily a myeloid differentiation antigen with initial expression at the very early stages of myeloid cells’ development and it is not expressed outside the hematopoietic system or on pluripotent HSCs. FLT3-ITD and NPM1^mut^ were found to have increased expression of CD33 in patients. Both antigens are expressed on proliferative blasts and LSCs and dual targeting of CD33 and CD123 may enhance treatment efficacy for AML [60]. Several approaches, such as monoclonal antibodies and CAR-T cells, have been suggested to target overexpression of CD33 and CD123 on AML blasts. Lintuzumab is a humanized IgG1 against CD33 that has shown only modest activity in clinical trials [61,62,63]. Due to its toxicity in the liver and normal hematopoietic cells, the Phase 3 trial was terminated [64], reflecting the fact that CD33 is largely expressed on myeloid lineage cells, including progenitor cells as well as on neutrophils, natural killer (NK) cells, a subset of B cells, and Kupffer cells in the liver [65].

Due to the short-term effect of αCD33, caused by its rapid internalization when it binds, bivalent antibodies conjugated to toxins have been developed to enhance CD33’s efficacy [64]. In 2017, after a decade of controversy, the Food and Drug Administration (FDA) authorized the use of GO, a humanized monoclonal αCD33 antibody, coupled with the cytotoxic chemical calicheamicin. When GO binds to CD33, it is internalized and calicheamicin is released, triggering DNA double-strand breaks that cause cell death.

Several trials have shown that augmenting the standard treatment with GO has had promising results. It is recommended that GO is added to the first cycle of standard 7+3 induction therapy, especially for CD33-positive AML NPM1^mut^ [12,66,67]. In the randomized AMLSG 09-09 study for NPM1^mut^ patients, CR rates were similar with and without GO, but relapse-free survival was prolonged to a clinically relevant extent in those patients who achieved a CR [68]. The inclusion of GO in intensive chemotherapy for AML NPM1^mut^ patients led to a notable decrease in NPM1^mut^ transcript levels throughout all treatment cycles, explaining this significant reduction in the relapse rates [35,36].

GO has been approved in conjunction with aggressive treatment for newly diagnosed AML patients [69]. GO was initially designed as a monotherapy with a single 9 mg/m^2^ dose for the first recurrence of CD33^+^ AML. Later, the dosage was adjusted to 3 mg/m^2^ on Days 1, 4, and 7 (or possibly only one dose on Day 1) of induction therapy and on Day 1 of consolidation therapy [12]. In the AML-SG study, a randomized trial was conducted to evaluate GO in combination with intensive induction and consolidation therapy in NPM1-mutated AML, as the high CD33 expression in AML with mutated NPM1 provides a rationale for the evaluation of GO in this AML entity. A higher mortality rate was observed in the GO group. More patients over the age of 70 died during induction therapy in the GO group than in the standard group [36].

In another study, the CR recovery rate was 13%, with a median recurrence-free survival of 6.4 months. The most common consequences were Grade 3 and 4 neutropenia (98%), thrombopenia (99%), and a few veno-occlusive disorders (0.9%). In up to 25–35% of patients with newly diagnosed or relapsed/refractory (R/R) AML, GO induced CR or CR with incomplete platelet recovery, especially in AML patients with a favorable prognosis [69]. GO also eliminated MRD in extensively treated individuals with a reduced response [70]. GO enhanced the 5-year OS and significantly reduced relapse rates in AML patients with NPM1^mut^ (*p* = 0·0028) [71]. However, combining GO with intensive CT for NPM1 AML patients in a randomized AMLSG 09-09 trial (NCT00893399) resulted in a high mortality rate [72]. Patient-administered GO had lower associated relapse rates compared with the standard treatment [72].

### 5.2. αCD123

Targeting CD123 has been suggested as an AML treatment; however, its efficacy was unsatisfactory, except in AML NPM1^mut^/or patients with a FLT3-ITD mutation. This is because these patients have an LSC population enriched in CD34^+^/CD38^−^ cells [73]. Using a naked antibody that targeted CD123 (CSL362) caused lysis of the AML cells that expressed CD123 due to the activation of the innate immune system [74]. However, in a clinical trial of 40 patients with R/R AML, only two patients had a clinical response [74].

Tagraxofusp (SL-401) is a recombinant αCD123 (IL-3 receptor alpha chain) antibody based on the diphtheria toxin. Following its attachment to CD123 and the inhibition of eukaryotic elongation factor 2 (eEF2), tagraxofusp was internalized. In patients with a blastic plasmacytoid dendritic cell neoplasm, a myeloid malignancy with high CD123 expression, SL-401 has led to clinical improvement [75]. To treat CD123-positive AML and MDS, SL-401 has also been used with azacitidine (AZA) or venetoclax/AZA. Three of the four previously untreated MDS patients and eight of the nine AML patients who had not received treatment before achieved CR. It is noteworthy that TP53 mutations were present in three responding MDS patients and two responding AML patients. When paired with venetoclax/AZA in CD123-positive R/R AML, pivekimab sunirine, another αCD123 antibody–drug combination that causes DNA alkylation, produced objective response rates (ORRs) of 51% with promising efficacy [76].

Using a mouse model that targeted both CD33 and CD123 via a bispecific conjugate, LSCs were eliminated through a T-cell-dependent mechanism both in vivo and in vitro [77]. This model suggests that AML NPM1^mut^ is initiated by CD123^+^ LSCs [77].

### 5.3. The Immune Checkpoint Inhibitors—Antibodies That Bind Programmed Cell Death-1 (αPD-1) Protein and Its Ligand (αPD-L1)

Checkpoint inhibitors targeting the programmed cell death-1 (PD-1) protein and its ligand (PD-L1), known as the PDx axis, have successfully achieved responses in a number of tumor types, including hematopoietic malignancies (reviewed in [78]). However, responses in AML have shown fewer promising results, perhaps reflecting their lower mutation burden. This may explain why recent ex vivo studies may provide evidence that supports the use of immunogenic strategies for AML NPM1^mut^ in the presence of an ICI.

A systematic review that compared 19 randomized clinical trials involving 11,379 patients with solid tumors including non-small cell lung cancer, renal cell carcinoma, and gastric urothelial carcinoma [79] showed that αPD-1 treatments (such as nivolumab and pembrolizumab) achieved better OS and PFS compared with αPD-L1 therapeutic antibodies, with no significant difference in safety profiles. The reason is thought to be that αPD-1 can bind PD-1 on T cells, blocking the binding of PD-1 to both PD-L1 and PD-L2 ligands on antigen-presenting cells (APC; cancer cells) at the same time. However the PD-L1 antibody can only bind PD-L1, so T cells may still be inhibited by the interaction between PD-1 and PD-L2, even when PD-L1 signaling is blocked (reviewed in [80]). The level of PD-L1 on APCs may affect the response of patients to αPD-L1 treatment, with PD-L1^low/negative^ patients having a compensatory increase in PD-L2 levels, making the difference in the treatment’s efficacy between αPD-1 and αPD-L1 exacerbated.

Expression of CD34/CD38/CD274 surface markers for LPC/LSCs was evaluated in 20/20 *NPM1^mut^*/*NPM1^wt^* AML patients’ samples via flow cytometry analyses. The LSC fraction showed a higher level of PD-L1 expression than the non-LSC fraction. The influence of αPD-1 antibodies on these antigen-specific immune responses and the formation of stem-like colonies were assessed as well. It is noteworthy that high PD-L1 expression in NPM1^mut^ patients was detected, especially in the leukemic progenitor compartment. This observation further supports the hypothesis that NPM1-directed immune responses might play an important role in tumor cell rejection, which tumor cells try to escape via the expression of PD-L1. The immunogenicity of neoantigens derived from NPM1^mut^ cells with higher PD-L1 expression constitutes promising target structures for individualized immunotherapeutic approaches.

In addition, we have previously shown that CD8^+^-specific LAA immune responses against Preferentially expressed Antigen in Melanoma (PRAME), WT-1, RHAMM, or NPM1^mut^ expressing LPCs/LSCs from AML patients with mutated or wild-type NPM1 were similar except for their response to NPM1^mut^ epitopes which were only seen in samples from NPM1^mut^ patients (vide supra). We also found that T cell-mediated anti-tumor responses from AML patients were enhanced by the presence of αPD-1 blocking antibodies [59]. Along with the expression of PD-1 on LSCs isolated from NPM1^mut^ patients, this suggests that treatment with αPD-1 antibodies combined with immunotherapeutic vaccine approaches could represent new treatment options for this biologically distinct group of patients [59,81].

The immune-modulating drug lenalidomide is a synthetic compound derived from structural modifications of thalidomide, a drug which was banned in the 1960s because of teratogenicity. Lenalidomide was released to market in 2004 and has been used to treat adults with multiple myeloma, smoldering leukemia, and AML. It has been shown to activate T cells and NKT cells, increase NKT cell numbers, and inhibit the production of pro-inflammatory cytokines such as interleukin-6 (IL-6) and tumor necrosis factor-alpha (TNF-α by monocytes. It effectively stimulates anti-tumor immunity, including anti-inflammatory, anti-proliferative, pro-apoptotic, and anti-angiogenicity [82]. Lenalidomide has been shown to achieve a 30% response rate as a first-line treatment in older newly diagnosed AML patients with limited treatment options and a poor prognosis [83].

The effects of lenalidomide are distinct from CT, hypomethylating agents, or kinase inhibitors, making lenalidomide an attractive agent for use in AML treatment, including in combination with existing active agents. Combinations of immunotherapeutic approaches are believed to increase antigen-specific immune responses against leukemic cells, including LPC/LSCs. In this study, we used a combination of LAA peptides with ICIs, αPD-1 (nivolumab), and lenalidomide to enhance the killing of LSC/LPCs ex vivo [82]. We used primary cells from AML patients and observed the T cell responses of all patients in vitro. Results using the combination of the ICI αPD-1 and the immune modulatory drug lenalidomide showed enhanced killing of LPCs/LSCs. The effect was greatest against NPM1^mut^ cells when the immunogenic epitope was derived from the mutated region of NPM1 [23]. These studies show that antigen-specific immune responses against leukemic cells as well as LPCs/LSCs are enhanced by combinations of immunotherapeutic approaches, especially when using a combination of LAA peptides, the αPD-1 antibody, and one further immunomodulating drug, providing an interesting opportunity to enhance the outcomes in future clinical studies.

We recently examined whether all-trans retinoic acid, AZA, αCTLA4, or lenalidomide, in combination with αPD-1, to determine whether this could further improve the destruction of leukemic cells and also LPCs/LSCs from AML patients (Greiner et al, in preparation) (Figure 2). We identified which LAA (WT-1, PRAME, or NPM-1) generated the strongest immune response (as measured by colony-forming immunoassays) and using samples from 20 AML patients with more than 90% leukemic blasts. AZA, in combination with αPD-1, had a pronounced effect on T cell activation and inhibition of LPC/LSC growth.

Other strategies for immunotherapy that can be used in combination with PDx need to be considered. One approach is the signal transducer and activator of transcription 5 (STAT5), which promotes PD-L1 expression by facilitating histone lactylation and driving immunosuppression in AML and thus may benefit from PD-1/PD-L1 immunotherapy. ICIs could block the interactions of PD-1 with PD-L1 and thereby enhance the reactivity of CD8^+^ T cells in the microenvironment when co-culture with STAT5-activated AML cells occurs [84].

**Figure 2 cancers-16-03443-f002:**
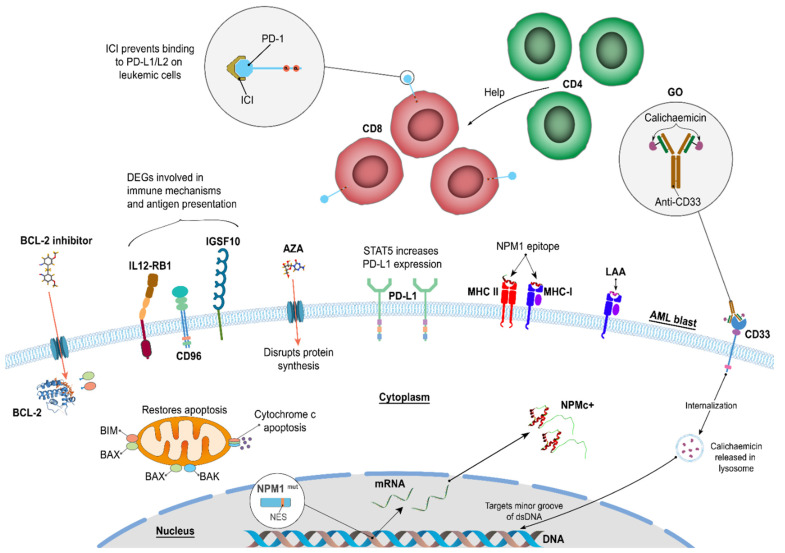
Combinations of immunotherapeutic treatments can use complementary strategies to enhance anti-leukemic responses. We have shown that these combinations can increase antigen-specific immune responses against leukemic cells as well as LPCs/LSCs [82,85]. Image generated using information from the following articles [5,6,7,8,35,36,38,39,46,52,60,67,68,69,70,71,72,81,84,85,86,87,88,89,90]. Aver et al. found that AZA in combination with αPD-1 resulted in improved OS and an encouraging response rate, particularly in hypomethylating agent (HMA)-naïve and salvage-1 patients [87]. A randomized Phase 3 study and a randomized Phase 2 study of AZA with or without PD-1 inhibitor in first-line elderly AML patients, and a randomized trial of a PD-1 inhibitor for the eradication of MRD in high-risk AML in remission have started. Clinical and immune biomarker-enriched trials are likely to yield further improved outcomes with HMA in combination with ICI therapies in AML, but this remains to be seen as more study results are made available, especially for AML NPM1^mut^ patients.

The results suggest that PD-1/PD-L1 in combination with NPM1mut-related approaches or other strategies such as STAT5 or HMA may have a notable impact on immunotherapeutic outcomes when used for the treatment of AML.

## 6. Venetoclax and Hypomethylating Agents

The aberration of DNA methylation is a common early event in the pathogenesis of AML, caused by genetic alterations in DNA methyltransferases (DNMTs) and Ten-Eleven-Translocation dioxygenases and leading to transcription dysregulation (reviewed in [88]). Elderly patients (≥65 years) with AML often respond poorly to induction chemotherapy [91]. This is mostly due to a higher frequency of adverse genomic features, increased resistance to treatments, partly caused by BCL-2 overexpression and also because of comorbidities, compromised organ function, and poor performance status. HMAs have been shown to upregulate silenced leukemia antigens and to promote the re-expression of endogenous retroviral elements. They exert inhibitory effects against cancer cells, which include stimulation of anti-tumor immunity in AML. In addition, the BCL-2 inhibitor venetoclax acts as a modulator of immunometabolism and potentiates adoptive NK cell immunotherapy. Both are not considered as standard immunotherapeutics, but their activity may make them appropriate for the stimulation of anti-tumor effects and circumvention of treatment resistance in older AML patients (>65 years) [17,92]. Recently, DiNardo et al. [39] showed that older treatment-naïve AML patients (>75 years) who were ineligible for intensive chemotherapy demonstrated a longer OS with increased remission rates when receiving AZA and venetoclax than with AZA alone. For newly diagnosed AML NPM1^mut^ patients who are unfit and/or elderly, the standard of care is a combination of venetoclax and HMAs such as AZA, which inhibits DNMTs by its incorporation into DNA or low-dose cytarabine (LDAC). Post-remission maintenance therapy with AZA has been shown to prolong MRD negativity, and may delay or prevent a relapse and improve OS in elderly patients with AML, including NPM1-mutated AML, independent of the initial MRD status [93]. Oral AZA treatment extended the duration of MRD negativity by 6 months vs. a placebo. Oral AZA was recently approved in the United States, Canada, and the European Union for the treatment of adult patients with AML in first remission following induction chemotherapy who are not able to complete intensive curative therapy (e.g., SCT).

The FDA-approved molecule venetoclax has been shown to target the NPM1 apoptotic pathway by inhibiting B cell leukemia/lymphoma-2 (BCL-2) [39,89,90]. The increased sensitivity to venetoclax may be due to an NPM1 mutant-primed impairment in mitochondrial function [94]. Venetoclax-based regimens are especially beneficial for AML NPM1^mut^ patients at first diagnosis and in the case of patients who are R/R to treatment. Regarding the effectiveness of venetoclax in AML NPM1^mut^, the human monocytic leukemia cell line THP-1, when transfected with a NPM1^mut^ containing vector, exhibits increased sensitivity to chemotherapies through the reduction in NF-κB activity and BCL-2/BAX expression [95]. High BCL2 protein levels are associated with improved outcomes in patients with R/R AML or untreated AML, according to predictive indicators for venetoclax sensitivity [96]. However, not every patient has experienced this. A plausible rationale could be that a FLT3-ITD or TP53 deletion in AML NPM1^mut^ patients leads to a mitochondrial malfunction. Moreover, as AML NPM1^mut^ patients age, IDH1/2 mutations develop, which impacts how well venetoclax-based therapy can work [97].

Single-agent HMA or LDAC regimens are often better tolerated and have lower treatment-related mortality rates than conventional CT [98]. With a median survival of 6–10 months, response rates to HMAs alone are modest (10–50%) [98,99]. Despite its efficacy, relapses within 18 months are a major obstacle of HMA due to drug resistance. One suggested mechanism is that HMA interferes with DNA methylation and, in the long term, causes the failure of cells to undergo cell cycle arrest [100].

The combination of HMA and venetoclax is suggested to maximize the efficacy of treatment, evading the resistance created through the use of a single agent. In a Phase 3 clinical trial (NCT02993523) involving venetoclax plus AZA [39], 66.7% of AML NPM1^mut^ patients experienced CR and incomplete remission, with the majority of the patients testing negative for detectable MRD. With a predicted two-year survival rate of 70%, the one-year survival rate for elderly individuals with AML NPM1^mut^ surpassed 80%. Usually, one or two cycles were required to generate the desired response with a favorable safety profile [39]. Similarly, venetoclax plus LDAC achieved CR and an incomplete CR of 89% and 78%, respectively, in AML NPM1^mut^ patients in the NCT02287233 and NCT03069352 trials [101,102]. The combination of venetoclax and HMA is considered to be advanced in AML treatment, and dose scheduling and further understanding of the mechanism of action and resistance remain to be explored [39].

## 7. Discussion

Our review has described treatments that are of particular relevance to AML NPM1^mut^ patients. These patients are unique, with a tendency to be over 35 years of age, have a high white cell count, and to have FLT3-ITD, a normal karyotype, upregulated HOX and HOX-associated genes, and low to absent CD34^+^ expression on their blasts. Representing one-third of all AML patients, AML NPM1^mut^ has become a separate entity in the WHO classification of AML, reflecting the patients’ ability to benefit from a targeted treatment strategy. Although the treatment of AML patients has changed little until recently [31], the AMLSG09-09 clinical trial showed that the addition of GO can extend relapse-free survival to a clinically relevant extent in CD33^+^ AML NPM1^mut^ patients who achieve a CR [36,68]. It is notable that most of the diseased cells in AML NPM1^mut^ patients are CD33^+^ and CD34^−^, even though LSCs are usually seen in the CD34^+^ and CD38^−^ fraction from AML patients [102]. This suggests that if NPM1^mut^ occurs in the LSC population, the LSCs are readily depleted by the immune system, or that NPM1^mut^ tend to occur in non-LSCs which are more easily seen by the immune response, leading to their destruction and the associated improved patient survival.

This is reflected by the association between NPM1^mut^ and improved survival in older patients (>70 years old) who have normal cytogenetics, independent of other molecular and clinical prognostic indicators, and in contrast to younger patients [103]. The haplotypes HLA-B*07, B*18, and B*40 are depleted in patients with NPM1^mut^, but when patients with NPM1^mut^ express one of these haplotypes, OS is increased [47]. The association of *NPM1^mut^* with a better prognosis for patients (reviewed in [104]) may be explained by more robust immune responses against leukemic cells in general and the LSC fraction in particular. Indeed, Schneider et al. showed that AML NPM1^mut^ patients harboring CD8^+^ responses against one of two predicted NPM1^mut^ peptides had a better OS that those without [45].

Recently, combinations of hypomethylating agents and the Bcl-2 inhibitor venetoclax gave impressive results in older patients with AML NPM1^mut^ compared with traditional standard-of-care regimens [105]. However, NPM1^mut^ patients with adverse cytogenetics should be treated the same as NPM1^WT^ patients with unfavorable cytogenetics who were linked to a lower 5-year OS rate [24].

Greiner et al. showed that CD4^+^ and CD8^+^ T cells from AML patients could respond to peptides from the mutated regions of NPM1 [44]. They found that AML NPM1^mut^ patients with CTL responses against the NPM1^mut^ peptide had better OS [47], and used LAAs as a specific method of stimulating T cell immune responses against LSCs and determining which combinations of immunotherapeutics could further enhance the killing of AML cells [82].

Elevated PD-L1 expression on AML NPM1^mut^ cells (a ratio of v1/v2, as determined by qPCR) is associated with poorer survival [106], suggesting that anti-PD-1 treatment could block this signaling pathway, enhance T cells’ killing of leukemic cells and cause an associated improvement in patient survival. In vitro, we showed that the immune checkpoint inhibitor αPD-1 can enhance killing by LAA-stimulated CTLs [46,59]. Notably, the effect was greatest when T cells from NPM1^mut^ patients were stimulated with the NPM1^mut^ peptide, suggesting that anti-PD-1 antibodies, such as nivolumab, could be used to treat AML NPM1^mut^. This may reflect the features of AML NPM1^mut^ that are especially associated with successful αPD-1 treatment of cancers including disease stability and a targetable mutation.

We recently examined whether αPD-1, all-trans retinoic acid (ATRA), AZA, αCTLA4 and/or lenalidomide were the best treatments to facilitate the destruction of autologous AML colonies in NPM1^mut^ and NPM1^WT^ patients [85]. The results were enhanced in patients with NPM1^mut^ compared with NPM1^WT^, with αPD-1 and AZA providing the most effective T cell-mediated reduction in AML cells in CFI assays. This concurs with our understanding of PDx treatments and when they work best. When CTLs recognize LAAs, their response includes the production of inflammatory cytokines such as IFNγ and the upregulation of PD-1 on their surface. This prevents excessive T cell responses and limits the immune response. The tumor cells concurrently upregulate PD-L1, inhibiting T cell responses against them [107]. The blockage of this PD-1/PD-L1 interaction can lead to enhanced tumor killing and improved patient survival rates (reviewed in [80]). However, results on the treatment of AML have been mixed, reflecting the features of NPM^WT^ that make AML a poor target for PDx therapies. These include rapidly progressing disease that generates a high tumor burden but a low mutational burden. However, PD-1 is regulated by DNA methylation [107], demonstrating a benefit for HMA and αPD-1 treatment in AML patients. The combination of venetoclax and HMAs has been shown to improve OS and CR compared with AZA alone in patients > 75 years of age with previously untreated AML who are ineligible for intensive chemotherapy [39].

It is clear that the development and progression of AML is closely linked to an impaired immune response. Leukemic cells manipulate the tumor microenvironment by creating a niche that directly promotes their survival and makes them resistant to immunotherapeutic strategies. Therefore, innovative strategies for immunotherapies need to be further developed and adapted to overcome the obstacles in the treatment of AML and especially AML NPM1^mut^ [108]. However, AML NPM1^mut^ offers several advantages over AML NPM^WT^, by virtue of their predominance of CD33^+^ diseased cells and an absence of CD34^+^, suggesting that the target population is not the more difficult to treat CD34^+^ LSC cells. The presence of NPM1^mut^ provides a unique target for treatment, whose immunogenicity can be enhanced by treatment strategies that are likely to be enhanced by αPD-1, HMAs, and immune modulators.

## 8. Conclusions and Future Developments

In the hunt for additional NPM1^mut^ peptide targets for future treatment strategies and, ideally, the identification of those that are naturally processed and presented, we could use mass spectrometry technologies that have been previously used to identify LAA-associated peptides processed and presented on MHC on CML and AML cells [107,109]. Similar strategies could be used to isolate peptides presented by MHC Class I and II molecules on CD34^−^CD33^+^ cells from AML NPM1^mut^ patients, especially those associated with improved survival. This would provide naturally occurring epitope(s) for future studies that could be used to enhance NPM1^mut^-directed immune responses. It may even lend credence to the development of NPM1^mut^ analog peptides, shown previously to stimulate enhanced immune responses against naturally occurring epitopes present on HLA-A*0201^+^ AML cells [54].

We have discussed the association between AML NPM1^mut^ and the unique features associated with it. These include normal cytogenetics and enhanced survival in patients > 70 years of age. This phenomenon appears to be associated with the capacity of NPM1^mut^ cells to stimulate a CD8^+^ T cell response against AML cells, leading to relapse-free survival in patients who achieve a CR. More recent studies have built on the standard-of-care treatment to show that the additional use of hypomethylating agents and Bcl-2 inhibitors can cause improved OS and CR in patients > 75 years of age who are not eligible for intensive chemotherapy. Future studies should examine other available treatment options and their efficacy in combination and the clinical trials that provide the proof of principle for a group of elderly AML patients who now have meaningful treatment options.

## Figures and Tables

**Figure 1 cancers-16-03443-f001:**
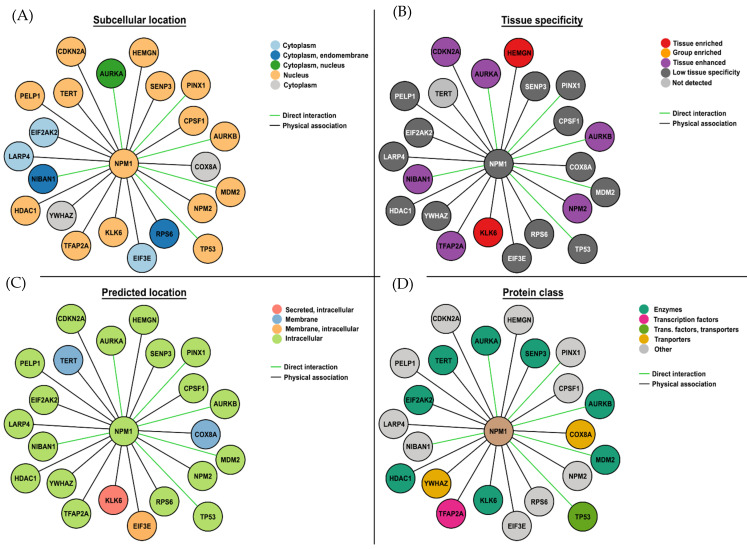
Direct interactions between wild-type NPM1 (NPM1^WT^) and proteins. The (**A**) subcellular localization, (**B**) tissue specificity, (**C**) predicted subcellular localization, and (**D**) protein class are indicated for each protein that NPM1^WT^ interacts with. Data taken from Protein Atlas (https://www.proteinatlas.org/; accessed on 18 June 2024) and drawn in Photoshop. As expected, most interactions are with nuclear proteins, most proteins are expressed in multiple tissues, intracellular locations, and enzymes based on direct interactions and proximity.

**Table 1 cancers-16-03443-t001:** Notable associated features of AML patients with NPM1^mut^ (based on [15,16]).

Associated Characteristics	AML NPM1^mut^	AML NPM1^WT^
Key information	Gatekeeper mutation, association with a specific subgroup of AML patients, has its own WHO subgroup	
Marker stability	NPM1^mut^ is a stable marker Observed again at relapse	
Age/sex	Associated with older AML patients > 35 years of age, de novo AML, and an increased frequency in females	More common in patients < 35 years of age.
Response to treatment	Good response to induction therapy	
Prognosis	Better prognosis in older but not younger AML patients; MRD status affects prognosis; NPM1^mut^/FLT3-ITD^−^ patients have a better prognosis than NPM1^mut^ FLT3-ITD^+^ patients	
Clinical features	Presents with high blast percentages, elevated white cell and platelet counts, and a high frequency of NK cells	
Karyotype	Normal karyotype	t(8;21), inv(16), t(15;17)
FAB subtype	FAB M1-M6; more often M4 and M5	FAB M0
Diseased cell phenotype	Restricted to myeloid cells Diseased cells are CD33^+^ and for > 90% of AML patients are CD34^−^ LSCs co-express CD96, IL12RB1	LSCs are CD34^+^CD38^−^
Mutations	FLT3-ITD (2 × more common); co-mutations include with *DNMT3A* > *FLT3-ITD* > tet methylcytosine dioxygenase 2 (*TET2*).	Biallelic CEPBA mutations occur
Associated gene expression	Upregulated HOX genes (A4, A5, A6, A7, A9, A10, B2, B3, B5, B6) and upregulated HOX-related genes (*PBX3* and *MEIS1*).	

DNMT3: DNA methyltransferase 3; FLT3-ITD: fms-related receptor tyrosine kinase 3-internal tandem duplication; LSCs: leukemic stem cells; NK: natural killer.

**Table 2 cancers-16-03443-t002:** Definition of AML-NPM1^mut^ risk groups by concurrent clinical, molecular, and cytogenetic indicators.

Risk	Molecular and Cytogenetic Indicators in AML NPM1^mut^ Patients
Poor	Partial tandem duplication of the mixed lineage leukemia *(MLL)* gene [25,26] and increased expression of the transcription factor ecotropic virus integration site 1 (EVI1) [27]; DNMT3A, MN1, BAALC, EGR-1, AF1q [28]; adverse cytogenetic abnormalities [29]
Intermediate	Isocitrate dehydrogenase 1 (nicotinamide adenine dinucleotide phosphate (NADP^+^)), soluble (IDH1), isocitrate dehydrogenase 2 (NADP^+^), mitochondrial (IDH2), and TET2 [30]; t(9;11)(p21.3;q23.3); MLLT3-KMT2A [31]; FLT3-ITD [12]; dependent on MRD status [12]
Favorable	NPM1^mut^ without any other genetic abnormality [12] or with a secondary type of mutation [32]; mutations in the transcription factor CEBPA indicates response to therapy [33,34]; concurrence with t(8;21), inv(16)/t(16;16), t(8;21) [12].

## Data Availability

No new data were created or analyzed in this study. Data sharing is not applicable to this article.

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
