# Peer review of "Immunotherapeutic Potential of Mutated NPM1 for the Treatment of Acute Myeloid Leukemia"

_cancers, 2024, doi:10.3390/cancers16203443_

Round 1
Reviewer 1 Report
Comments and Suggestions for Authors
In this review, Greiner and colleagues write about NPM1mut AML, the immunogenic capacity of this AML subtype, and the therapeutic implications arising thereof.
The review is generally well-written, and the figures are beautifully arranged. The immunological mechanisms are detailed, and treatment implications are well explained. However, I believe that the current treatment strategies are insufficiently explained, sometimes misleading, and—at least to my knowledge—sometimes incorrect. The authors should place more focus on that. In detail, the following issues need to be addressed.
- The authors state that NPM1 mutations are only associated with a favorable prognosis in older but not younger patients and base that on a single real-world registry. While the Swedish registry is extensive and can be discussed, there are other data as well. The ELN2022 is established in younger patients with high-dose chemotherapy and has NPM1 as a favorable marker. Also, reports from randomized trials show the opposite (DOI: 10.1200/JCO.2014.58.0571).
- The authors state in lines 126-127 that adverse risk cytogenetics in NPM1+ AML impacts the favorable prognosis and should be treated like adverse-risk AML. This is correct and stated in the ELN2022. Table 2 shows different risk groups based on cytogenetic and molecular co-occurrences. However, the ELN 2022 only states adverse risk cytogenetics and FLT3-ITD as prognostically relevant co-occurrences. Also, it has been shown that co-occurring adverse-risk mutations do not alter the prognosis (DOI: 10.1038/s41375-023-02016-6). The authors should discuss that in more detail and comment on it and correct the table.
- The chapter with the AML treatment strategies should be written more precisely and include the currently accepted concepts about NPM1-MRD as outlined in the ELN2021 MRD guidelines (doi: 10.1182/blood.2021013626.). As far as I can see, it must be referenced or discussed. I also strongly disagree with Table 3 here. The authors state, “1st line CT: AlloHSCT indicated due to intermediate or poor-risk genetic features (n=27)”. This is against the ELN guidelines, where NPM1 in 1st line gets an allo-Tx candidate when it co-occurs with adverse risk CG and FLT3-ITD but not with other genetics. The authors should comment on this. Also, the role of MRD and MRD failure should be discussed in more detail here. Recently, Othman and colleagues showed that MRD- after two high-dose cycles does not profit from alloTx, even if they have an FLT3-ITD or other co-occurring mutations; however, MRD+ after two cycles did (doi: 10.1182/blood.2024024310). These things should be discussed, and the current treatment strategy should be clarified, maybe even with a flow diagram.
- For GO, the authors stated the 9mg dose, which was used during its first licensing period. They should emphasize that it is much lower now; otherwise, it could be misleading. Also, they should mention that in the AMLSG trial, older patients did not profit because of increased early mortality rates.
- In the discussion, the authors state that “The standard treatment protocol for newly diagnosed NPM1mut AML patients who are unfit and/or elderly (>60 years), is a combination of venetoclax and HMAs such as AZA which inhibits DNMTs by its incorporation into DNA or low-dose cytarabine (LDAC).” I strongly disagree with this statement. 60 years is not a cut-off for elderly not eligible for high-dose chemotherapy. Particularly for NPM1 fav-risk AML, who probably will not need alloTx, the general concept is more to raise the age limit for high-dose chemotherapy and not bring them into the indefinite VEN/AZA regimen.
- Following point 5, the concept of high-dose chemotherapy and oral AZA maintenance (see posthoc analyses of Quazar from Döhner H et al.) is not explained or discussed.
In conclusion, the manuscript shows a good overview on immunological parameters of AML NMP1+ with reference to treatment strategies therein. However, the parts about the current clinical practice in this disease type are not presented comprehensively, sometimes unclear, and sometimes incorrect. This should be changed before the manuscript can be accepted.
Reviewer 2 Report
Comments and Suggestions for Authors
Overall, this review is not clear, schemas and figures to sum-up the treated arguments are lacking,and paragraphs often lack a final point or a proper analysis of the treated argument.
I also find this review inaccurate; I reported some examples:
- Author stated that the most frequent NPM1 mutation is mut A followed by mut D, but the second most frequent is the mut B.
- Overall, the table 1 is not clear: if the message is the difference between AMLmut and wt, it would be convenient to insert a first column with the characteristic evaluated (i.e. in the second row “Age” and fill out all the part (also in NPM1 wt). Moreover: it is important to highlight that as NPM1mut is a stable marker, it is suitable for MRD monitoring; make explicit “WCC”.
- In the line 271 it is repeated that GO is a conjugate of CD33 antibody and cytotoxin calicheamicin: it was already described 2 lines above.
- There is a paragraph reporting data of treatment with this Lenalidomide, but this drug is barely described, just defining it as an immune-modulating drug.
- Line 421: the THP1 cell line does not harbor NPM1 mutation.
Many sentences are confused and not clear (for example: AML NPM1mut were not found to be associated with event-free survival, OS or probability of relapse at 60 months post-diagnosis but because NPM1mut were significantly associated with normal karyotypes and the presence of FLT3-ITD (Table 2) there appears to be a trend towards improved survival in patients in the intermediate risk group without 132 FLT3-ITD but with NPM1mut (p=0.05) [17].)
Authors suggest that in NPM1mut AML, LSC population is readily depleted by the immune system, or that NPM1mut occur in non-LSCs, based on the fact that “Most diseased cells in NPM1mut AML patients are CD33+ and CD34-; LSC are usually seen in the CD34+CD38- cell fraction”. However, to
date it is well established that LSCs are not only defines as CD34+CD38- , but also by multiple other genetic and functional features (for example with LSC17 signature, ROS low content, etc. ). Thus, this assumption of the author is largely incorrect or at least not supported to date, with the data
presented here.
Chapter 3 is very confusing, mixing chemotherapy and alloHSCT, without resulting in an analysis of the best treatment option in NPM1mut patients.
Chapter 4 potentially would have great interest, reporting T-cell response against HLA-A2-restricted peptides originated from NPM1mut AML. However, authors only described their own work, without discussing other papers relevant in the field (such as Liso A, Colau D, Benmaamar R,
De Groot A, Martin W, Benedetti R, et al. Nucleophosmin leukaemic mutants contain C-terminus peptides that bind HLA class I molecules. Leukemia. 2008;22:424–6. Narayan R, Olsson N, Wagar LE, Medeiros BC, Meyer E, Czerwinski D, et al. Acute myeloid leukemia immunopeptidome reveals HLA presentation of mutated nucleophosmin. PLoS ONE. 2019;14:e0219547. van der Lee DI, Reijmers RM, Honders MW, Hagedoorn RS, de Jong RC, Kester MG, et al. Mutated nucleophosmin 1 as immunotherapy target in acute myeloid leukemia. J Clin Investig. 2019;129:774–85. Kuzelova K, Brodska B, Fuchs O, Dobrovolna M, Soukup P, Cetkovsky P. Altered HLA class I profile associated with type A/D nucleophosmin mutation points to possible anti-nucleophosmin immune response in acute myeloid leukemia. PLoS ONE. 2015;10:e0127637.) limiting the relevance of this analysis.
Moreover, the latter part of the chapter (“Samples from 13/15 (87%) NPM1mut AML patients showed immune responses against at least one LAA and 9/15 (60%) against all four LAAs. Etc. In NPM1WT patients no responses were found against the NPM1mut epitope, but responses against the other LAAs were comparable to those of NPM1mut patients, although immune responses varied slightly depending on the antigen and the patient.) is totally uninformative regarding the immune response against LAA of NPM1mut vs NPM1wt AML patients.
Chapter 6 does not include immunotherapy, as the title of this review suggests.
no
Round 2
Reviewer 2 Report
Comments and Suggestions for Authors
see attached file

some paragraphs should be semplified and clarified
Round 3
Reviewer 2 Report
Comments and Suggestions for Authors
Authors have improved the clariry of this review.
Comments on the Quality of English LanguageEnglish language is now improved but still require minor editing.